# Exploring the Optoelectronic Properties of D-A and A-D-A 2,2′-bi[3,2-*b*]thienothiophene Derivatives

**DOI:** 10.3390/molecules27238463

**Published:** 2022-12-02

**Authors:** Levi Gabrian, Gavril-Ionel Giurgi, Ioan Stroia, Elena Bogdan, Andreea Petronela Crişan, Niculina Daniela Hădade, Ion Grosu, Anamaria Terec

**Affiliations:** Department of Chemistry and SOOMCC, Faculty of Chemistry and Chemical Engineering, Babes-Bolyai University, Cluj-Napoca, 11 Arany Janos, 400028 Cluj-Napoca, Romania

**Keywords:** thienothiophene, fluorescence, π-conjugated systems, electron-acceptor groups, DFT calculations

## Abstract

The synthesis of some novel donor-acceptor and acceptor-donor-acceptor systems containing a 2,2′-bi[3,2-*b*]thienothiophene donor block and various electron-accepting units is described alongside their photophysical properties studied using electrochemistry, optical spectroscopy and theoretical calculations. The obtained results show that the energy levels can be modulated by changing the strength of the acceptor unit. Among the three investigated end-groups, 1,1-dicyanomethylene-3-indanone exhibited the largest bathochromic shift and the lowest band gap suggesting the strongest electron-withdrawing character. Moreover, the emissive properties of the investigated systems vary greatly with the nature of the terminal group and are generally lower compared to their precursor aldehyde derivatives.

## 1. Introduction

Among the aromatic-fused 5,5 heterocyclic systems with one heteroatom in each ring, fused bithiophenes stand out due to their versatile properties and vast applications [1,2,3,4,5,6,7]. There are four possible arrangements of the heteroatoms ([Fig molecules-27-08463-ch001]) with stable aromatic structures **I**–**III**, while compounds with a core structure **IV** showing a S(IV) atom are not stable and could not be isolated.

By far, [3,2-*b*]thienothiophene ([3,2-*b*]-TT, **II**) is the most investigated [7] as its derivatives revealed a wide array of applications. Thus, researchers have found it useful in medicinal chemistry as antibacterial agents [8,9], powerful inhibitors of platelet aggregation [10] or in glaucoma treatment [11], as materials with liquid crystal behavior [12,13] or non-linear optical properties [14,15] and in polymer chemistry as conducting (co)polymers [16,17,18].

Relevant applications of [3,2-*b*]-TT derivatives are also connected to converting solar energy into electricity. Thus, they were employed for the fabrication of efficient dye-sensitized solar cells (DSSCs) [19,20,21,22] or classical organic solar cells (OSCs). In the latter, they were successfully used as building blocks in donor molecules [23,24,25,26,27,28], as non-fullerene acceptors [29,30,31,32,33,34] or were incorporated in monomers that, upon (co)polymerization reactions, lead to macromolecular products used as active materials in OSCs fabrication [35,36,37,38,39,40,41]. Moreover, it was reported that replacing a thiophene ring or a bithiophene unit either in acceptors (**A**) (e.g., [29]) or in donors (**D**) (e.g., [24]) with a [3,2-*b*]-TT moiety indicated an improvement in the efficiency of the OSCs exhibiting the modified donors and/or acceptors.

Starting from the outstanding properties of [3,2-*b*]-TT derivatives, we considered it of interest to expand our research in the field of molecules with exciting optoelectronic properties [42,43,44,45,46] and to investigate some novel 2,2′-bi[3,2-*b*]thienothiophene derivatives belonging to **D-A** (**V**) or **A-D-A** (**VI**) systems ([Fig molecules-27-08463-ch002]).

## 2. Results and Discussion

### 2.1. Synthesis

The first step to accessing the target compounds of types **V** and **VI** was building the 2,2′-bi[3,2-*b*]thienothiophene skeleton through the homo-coupling reaction of 3,6-dialkyloxy[3,2-*b*]thienothiophenes **1** and **2**, as previously reported [47]. We started with the synthesis of tetrakis(methyloxy) derivative **3** (Figure 1) using a procedure already reported in the literature employing 3,6-dimethoxy[3,2-*b*]thienothiophene **1** [48]. As compound **3** and its derivatives showed low solubility in usual solvents, replacing the methyloxy (compound **1**) with the hexyloxy (compound **2**) groups was mandatory to ensure good processability for the structure and properties investigations of the targeted new derivatives with 2,2′-bi[3,2-*b*]thienothiophene. The homocoupling of **2**, leading to 3,3′6,6′-tetrakis(hexyloxy)-2,2′-bi[3,2-*b*]thienothiophenes **4** (Figure 1), was performed in conditions similar to those used for the synthesis of **3**. In the next step, the formylation of **4** through a classical Vilsmeier-Haack reaction afforded either mono (**5**) or diformyl (**6**) derivatives (Figure 2), which were isolated as major products by modulating compound **4** to the *N,N′*-dimethylformamide (DMF) molar ratio.

Monoaldehyde **5** was reacted with malonodinitrile, *N*-ethylrhodanine and 1,1-dicyanomethylene-3-indanone, respectively, and gave good yields of the **D-A** compounds **7**–**9** (Figure 3). The reaction of dialdehyde **6** with 1,1-dicyanomethylene-3-indanone failed, but its Knoevenagel condensation with malonodinitrile and *N*-ethylrhodanine successfully led to **A-D-A** derivatives **10** and **11**, respectively (Figure 4).

### 2.2. Optical Properties

The physicochemical properties of the symmetrical and unsymmetrical systems based on [3,2-*b*]-TT **4**–**11** were investigated by UV-vis and fluorescence spectroscopy. The absorption properties have been studied in diluted CHCl_3_ (except for compound **8** that was solved in dichloromethane as it decomposed in chloroform) solutions (*ca* 10^−5^ M), as well as in thin films spun-cast on glass from chloroform solutions for the target **(A-)D-A** systems **7**–**11**. The corresponding data are gathered in Table 1. The UV-vis spectra (Figure 1) of all synthesized compounds reveal similar absorption features, with broad, intense absorption bands (λ_max_) in the visible region (435–600 nm) assigned to π-π* electron transitions between the highest occupied molecular orbital (HOMO) and the lowest unoccupied molecular orbital (LUMO), as the TD-DFT calculations revealed (*vide infra*), and less intense bands in the UV region (250–420 nm) ascribed also to π-π* transitions between orbitals separated by higher energies. The impact of the electron-accepting ability of the end-capping groups (dicyanovinylene, rhodanine and indanone derivatives, respectively) in the **D-A** and **A-D-A** systems on the electronic properties is observed by inspecting the trend of λ_max_ values (Table 1), as all the absorption maxima are red-shifted compared to that in parent [3,2-*b*]-TT compound **4** and aldehydes **5** and **6** as well, following the decrease of the optical band gap. Comparison the UV-vis spectra of the unsymmetrical **D-A** systems (**7** and **8**) and those of symmetrical **A-D-A** systems (**10**, **11**) supports the aforementioned effect, with significant bathochromic shifts observed in the case of the latter.

Moreover, as outlined in Table 1, the molar absorption coefficient (ε) of the low energy absorption bands of the **A-D-A** systems (79.8–98 × 10^3^ M^−1^cm^−1^) is higher than that of the unsymmetrical counterparts (62.5–61.7 × 10^3^ M^−1^cm^−1^), suggesting a better ability of light harvesting in the symmetrical systems. On the other hand, among the unsymmetrical systems, the indanone derivative **9** has the best light-harvesting ability (ε = 89.3 × 10^3^ M^−1^cm^−1^), higher than that of the **A-D-A** derivative **10** bearing dicyanovinylene as accepting units and very similar to compound **11** that has rhodanine moieties as terminal groups.

A comparison of the spectra in solution and neat film reveals two different behaviours. For compounds containing malonodinitrile and rhodanine as the acceptor unit (**7**, **8**, **10**, **11**), a broadening and a red-shift of the band could be observed that is commonly observed in formation of *J*-aggregates, while in the case of derivative **9** bearing an indanone derivative as the accepting moiety, a blue-shift of the absorption maximum band could be seen that occurs in *H*-aggregates [49]. The estimated solid-state band gap of the materials taken from the red-edge onset of the thin films varies between 1.92–1.53 eV and follows the same trend with the optical gap in solution (2.08–1.71 eV).

The fluorescence emission properties and the fluorescence quantum yield (QY) of the compounds have been measured in diluted chloroform (dichloromethane for **8**) solutions. Surprisingly, compound **9** showed no fluorescence. The fluorescence spectra are presented in Figure 2 and the emission characteristic data are summarized in Table 2.

As shown in Figure 2, the photoexcitation of precursor aldehyde derivatives **5** and **6** at λ_exc_ = 434 nm and λ_exc_ = 450 nm, respectively, provides an emission maximum near 500 nm related to a fluorescence quantum yield of 77% and 18%, respectively, relative to fluoresceine as the standard reference.

Interestingly, compared to precursor derivatives, the **D-A** and **A-D-A** systems also exhibit emissive properties (see Table 2), with a pronounced red-shift of the emission maximum and a significantly decreased QY.

A comparison between the mono and disubstituted [3,2-*b*]TT derivatives highlights that **A-D-A** systems **10** and **11** show poor photoluminescence properties. It is also worth mentioning that both rhodanine-based **D-A** and **A-D-A** derivatives (**8** and **11**) display very low fluorescence QY (<1%) compared to those bearing dicyanovinylene as the electron-withdrawing group (10% in the case of derivative **7** and 6% in case of derivative **10**).

Lifetime measurements for compounds **5**–**11** and estimation of the radiative (k_r_) and non-radiative (k_nr_) decay constants (Table 3) show that non-radiative processes are overwhelming in compounds **8**–**11**.
(1)$$k_{r}=\frac{\mathrm{QY}}{\tau }$$
(2)$$k_{\mathrm{nr}}=\frac{1-\mathrm{QY}}{\tau }$$
where QY is the experimentally determined quantum yield and τ represents the average lifetime of the compound.

### 2.3. Electrochemical Properties

To study the electrochemical properties of the synthesized compounds **7**–**11**, cyclic voltammetry (CV) measurements were carried out in chloroform using Bu_4_NPF_6_ as a supporting electrolyte. Cyclic voltammograms of **A-D-A** systems **10** and **11** (Figure 3) reveal reversible oxidation and reduction waves that suggest the formation of the corresponding stable cation and anion radicals. Unsymmetrical **D-A** derivatives **7**–**9** are reversibly oxidized and undergo irreversible reduction processes, as shown in Figure 3. In the case of compound **7**, the oxidation process is not fully reversible; in the reverse scan, a cathodic wave of weak intensity can be observed. This feature may be the result of the reduction product formed by the coupling of radical-cation [49].

As depicted in Table 1, the nature and number of the electron-acceptor group exerts a noticeable effect on both the anodic and cathodic peak potentials, which suggests that the HOMO and LUMO are delocalized over the entire molecule.

The electrochemical gap (E_g_) estimated from the onset of the redox processes has the same trend as observed in the UV-vis spectroscopy experiments (see Table 1). Thereby, in the **D-A** series, the more electron-withdrawing group (EWG) from dicyanovinylene (**7**) to 1,1-dicyanomethylene-3-indanone (**9**) causes a decrease of the band gap by 0.62 eV. The same behaviour is observed in the **A-D-A** systems when the dicyanovinylene (**10**) is replaced with *N*-ethyl rhodanine (**11**).

### 2.4. Theoretical Investigations

To have a clearer idea of the impact of the structure on the optoelectronic properties of the obtained **D-A** and **A-D-A** molecules, we performed density functional theory (DFT) calculations at the B3LYP-D3/Def2-TZVP level of theory [51,52,53] using the Gaussian 09 package [54] (see ESI for details). First, the geometries of compounds **7**–**11** were optimized without any symmetry constraint and with the convergence criteria set to tight. Moreover, to mimic the experimental conditions as much as possible, calculations were carried out in chloroform using the PCM model, except for compound **8**, which was solvated in dichloromethane. After the optimization step, all structures were found to be fully coplanar (Appendix A), which is not so surprisingly since there are no steric repulsions between the building blocks of each compound (i.e., the two TT units and acceptor part(s)) and also due to their tendency to facilitate conjugation.

With the optimized geometries in hand, we further performed time-dependent density functional theory calculations (TD-DFT) at the same level of theory and in the appropriate solvent (see ESI for details about the method) in order to find out the origins of the experimentally observed absorption bands. The theoretical UV-vis spectra agree well with the experimental results, with their comparison depicted in Figure 4, along with the most relevant transitions in the 300–1000 nm range and with an oscillator strength higher than 0.15. According to the TD-DFT results, in all cases, the lowest-energy absorption (and the most intense band at the same time) observed in UV-vis spectra is due to transition from the ground state (S_0_) to the first excited electronic state (S_1_), i.e., S_0_→S_1_, which involves the electron density moving from the HOMO to the LUMO with a probability of ~100%. Lower-intensity absorption bands also appear in the calculated spectra pertaining to transitions to higher excited electronic states. For example, the second absorption band (*ca.* 387 nm) in **7** can be associated with a combination of HOMO-1→LUMO and HOMO-2→LUMO transitions, while in **10**, a HOMO-2→LUMO transition is mainly responsible for the band around 418 nm. One should note that the S_0_→S_2_ transition is forbidden in all cases, except for compound **9**, where it is characterized by a low oscillator strength.

Concerning the low-energy absorption bands, their theoretical absorption maxima ranges from 546 nm to 645 nm within the **D-A** series and from 604 to 648 nm in the **A-D-A** type compounds, depending on the type and number of EWGs attached on the bi-TT core. Moreover, the theoretical bandgap (HOMO-LUMO) (E_gap_, Table 1) trend follows the experimental one and can be further explained by inspecting the energy levels of the frontier molecular orbitals (FMOs) for the individual EWGs and for the bi-TT central unit, respectively (Appendix A). More precisely, the InOCN has the lowest lying LUMO among all EWGs and contributes considerably to the lowering of the hybrid LUMO of molecule **9**. Interestingly, the rhodanine acceptor moiety keeps the E_gap_ of both the **D-A** (**8**) and **A-D-A** (**11**) systems below the E_gap_ of DCV counterparts (i.e., **7** and **10**), despite its inability to decrease LUMO eigenvalues as DCV does. The explanation is that the rhodanine part decreases the HOMO values with only 0.05 eV (in **8**) and 0.1 eV (in **11**) relative to parent dimer **5**, which is less than DCV moiety, which considerably decreases the HOMO in 7 and **10** (with 0.2 eV and 0.5 eV, respectively) relative to the same parent [3,2-*b*]-TT central unit.

As HOMO and LUMO have the strongest impact on the optoelectronic properties of our studied compounds, it is useful to depict their spatial distribution in order to obtain more information about the nature of the electronic transitions. On this note, careful inspection of the wave functions shows a distribution of HOMO over the entire molecule in all **(A-)D-A** motifs, with a lower coefficient on the acceptor moiety in the case of the DCV- and InOCN-based compounds. Similarly, LUMO is delocalized over both the donor and acceptor units within the **A-D-A** molecules, but with a considerably higher coefficient on the acceptor part in the case of the **D-A** counterparts (Figure 5). This orbital distribution suggests that the HOMO→LUMO transition within the **D-A** series can be regarded as a combination of local excitation (LE) and internal charge transfer (ICT), while in the case of the **A-D-A** counterparts, only local transitions take place. To further prove these statements, we plotted the electron density difference between first excited state and ground state for each molecule (Figure 6). Upon excitation, the electron density moves from the yellow region to the blue one; thus, in the case of compounds **7**–**8**, the S_0_→S_1_ has a mixed character of ICT from uncapped TT moiety toward the EWG and LE mainly within the other half of the [3,2-*b*]-TT core and also within the EWG, especially in compound **8**. On the other hand, in compound **10** and **11**, only LE takes place. The absence of ICT in our **A-D-A** molecules is most likely due to a few cumulative causes: (*i*) the absence of a π-spacer between **D** and **A** that could facilitate the localization of the HOMO on the donor side and the LUMO on the acceptor moiety (i.e., charge separation); (*ii*) the presence of two acceptor moieties, which pull the electron density in opposite direction and do not allow its polarization toward one EWG; and (*iii*) the imbalance between the donation ability of the [3,2-*b*]-TT platform and accepting ability of EWGs.

## 3. Materials and Methods

### 3.1. General Data

^1^H NMR (400 or 600 MHz) and ^13^C NMR (100 or 150 MHz) spectra were recorded in CDCl_3_ or CD_2_Cl_2_ at room temperature using the solvent line as a reference.

Thin layer chromatography (TLC) was conducted on silica gel 60 F_254_ TLC plates. All plates were visualized by UV irradiation at 254 and 365 nm.

Solvents were dried and distilled under argon using standard procedures. Chemicals were purchased from TCI Chemicals, and Alfa Aesar and Sigma Aldrich and were used without further purification.

Melting points were determined with a Kleinfeld apparatus and are uncorrected.

HRMS were recorded using an LTQ XL OBITRAP mass spectrometer equipped with ESI/APCI sources. UV-vis optical data in solution and in films were recorded with an UV-vis 1900 Shimadzu spectrometer. Fluorescence spectra were recorded on a JASCO FP-8300 spectrofluorometer using glass cuvettes (1 cm). Solutions for UV-vis, CV, HRMS and fluorescence measurements were prepared in HPLC grade solvents (dichloromethane, chloroform, acetonitrile).

Films were obtained from chloroform solutions using a classical spin coater.

Fluorescence quantum yields were calculated using fluoresceine (in 0.1 N NaOH, QY = 0.95) or rhodamine B (in ethanol, QY = 0.7) as standards.

Fluorescence lifetime measurements were performed on a MicroTime200 time-resolved confocal fluorescence microscope system (PicoQuant, Berlin, Germany), described in detail elsewhere [55]. The excitation was provided by picosecond diode laser heads operating at selected wavelengths (405 nm for **5** and **6**; 520 nm for **7, 8, 10, 11**; and 640 nm for **9**) and at a 40 MHz repetition rate. The signal was collected from the compounds in a CHCl_3_ solution with a UPlanSApo 60 ×/NA = 1.2 water immersion objective and filtered thereafter with a 50 μm pinhole and adapted long-pass emission filters.

Cyclic voltammetry (CV) measurements were carried out with a Biologic SP-150 potentiostat using a three-electrode cell equipped with a platinum electrode, a calomel reference electrode (SCE) and a platinum wire counter electrode. All experiments were performed in 0.10 M Bu_4_NPF/CHCl_3_ at 100 mV/s, Pt electrode references, SCE.

### 3.2. Procedure for the Synthesis of **4**

Under stirring and in an argon atmosphere, *n*-butyllithium (1.5 M in hexane) (1.83 mL, 2.75 mmol) was added dropwise over 10 min at −78 °C to a solution of 3,6-bis(hexyloxy)thieno[3,2-*b*]thiophene (850 mg, 2.5 mmol) in dry THF (65 mL) and the stirring was continued for another 2 h at the same temperature. Then, CuCl_2_ (370 mg, 2.75 mmol) was added in one portion and the reaction mixture was gradually warmed up to room temperature and then stirred overnight. After adding water (30 mL) and triethylamine (10 mL), the mixture was extracted with dichloromethane (3 × 30 mL) and the combined organic layers were dried over magnesium sulfate. After filtration, the solvent was removed under reduced pressure. The crude product was purified with silica gel gradient column chromatography (elution system started with pentane/CH_2_Cl_2_ = 2/0.2 and ended with pentane/CH_2_Cl_2_ = 2/0.4) to afford a yellow solid (410 mg, 0.60 mmol, 48%).

### 3.3. Procedure for the Synthesis of **5** and **6**

A solution of **4** (85 mg, 0.125 mmol) in dry 1,2-dichloroethane (10 mL) was purged with argon and cooled to 0 °C. Then, DMF (**5**: 15 μL, 0.19 mmol; **6**: 82 μL, 1.06 mmol) and phosphoryl chloride (**5**: 13 μL, 0.14 mmol; **6**: 82 μL, 0.88 mmol) were slowly added and the reaction mixture was additionally stirred for 1 h at 0 °C. Then, the reaction mixture was heated under stirring at 60 °C for 18 h. Afterwards, the mixture was poured into a NaOAc solution (20 mL, 1 M) and vigorously stirred for 2 h at rt. The final solution was extracted with dichloromethane (3 × 30 mL) and the combined organic layers were dried over magnesium sulfate. The solvents were removed under reduced pressure and the crude product was purified by silica gel gradient column chromatography (elution system started with pentane/CH_2_Cl_2_/Et_3_N = 2/1/0.02 and ended with pentane/CH_2_Cl_2_ = 2/1.5) to afford an orange solid for **5** (58 mg, 0.08 mmol, 66%) or a red solid for **6** (140 mg, 0.19 mmol, 86%).

### 3.4. Procedure for the Synthesis of **7** and **10**

Malononitrile (**7**: 47 mg, 0.71 mmol; **10**: 107 mg, 1.62 mmol) and triethylamine (0.1 mL) were subsequently added to an argon purged solution of **5** (50 mg, 0.07 mmol) or **6** (60 mg, 0.08 mmol) in toluene (10 mL) and the reaction mixture was stirred at room temperature for 18 h. Then, the solvent was removed under reduced pressure, the crude product was triturated with ethanol, and the precipitate was filtered off to afford a metallic green solid (**7**: 44 mg, 0.06 mmol, 82%; **10**: 65 mg, 0.08 mmol, 96%).

### 3.5. Procedure for the Synthesis of **8** and **11**

3-Ethyl-2-thioxothiazolidin-4-one (**8**: 41 mg, 0.26 mmol; **11**: 171 mg, 1.06 mmol) and piperidine (**8**: 0.35 mL; **11**: 0.5 mL) were subsequently added to an argon-purged solution of **5** (60 mg, 0.08 mmol) or **6** (130 mg, 0.18 mmol) in toluene (**8**: 20 mL; **11**: 50 mL) and the reaction mixture was stirred at 70 °C for 20 h. Then, the solvent was removed under reduced pressure and the crude product was purified by silica gel column chromatography (elution system for **8**: pentane/CH_2_Cl_2_/Et_3_N = 2/1/0.02; for **11**: toluene/CH_2_Cl_2_ = 2/0.2). Pure samples were obtained with further purification by crystallization through slow evaporation from a 2/1 mixture of dichloromethane_/_ethanol for **8**, and in the case of **11**, by precipitation with ethanol from its solution in chloroform. The products were obtained as metallic green solids (**8**: 40 mg, 0.05 mmol, 55%; **11**: 135 mg, 0.13 mmol, 75%).

### 3.6. Procedure for the Synthesis of **9**

2-(3-Oxo-2,3-dihydro-1*H*-inden-1-ylidene)malononitrile (33 mg, 0.17 mmol) and triethylamine (0.4 mL) were added subsequently to an argon-purged solution of **5** (40 mg, 0.06 mmol) in toluene (30 mL) and the reaction mixture was stirred at 60 °C for 3 h. The solvent was removed afterwards under reduced pressure and the crude product was purified by silica gel column chromatography (elution system: pentane/CH_2_Cl_2_/Et_3_N = 4/5/0.04). Further purification by crystallization from a 2/1 mixture chloroform/ethanol afforded a metallic green solid (25 mg, 0.03 mmol, 57%).

3,3’,6,6’-tetrakis(hexyloxy)-2,2’-bithieno[3,2-*b*]thiophene **4**: yellow solid; mp: 96–97 °C; 410 mg, 0.60 mmol, 48%; R_f_ = 0.4 (pentane/CH_2_Cl_2_ = 2/0.2). ^1^H-NMR (400 MHz, CDCl_3_) δ (ppm): 0.89–0.93 (overlapped peaks, 12H), 1.33–1.38 (overlapped peaks, 16H), 1.45–1.55 (overlapped peaks, 8H), 1.80–1.92 (overlapped peaks, 8H), 4.07 (t, 4H, *J* = 6.5 Hz), 4.32 (t, 4H, *J* = 6.5 Hz), 6.19 (s, 2H). ^13^C-NMR (100 MHz, CDCl_3_) δ (ppm): 14.20, 14.21, 22.75, 25.75, 25.82, 29.22, 30.13, 31.69, 70.76, 72.60, 97.46, 118.22, 127.60, 128.01, 144.86, 150.54. HRMS (ESI): *m*/*z* calcd for [C_36_H_54_O_4_S_4_ + H]^+^: 679.2978; found: 679.2966.

3,3’,6,6’-tetrakis(hexyloxy)-2,2’-bithieno[3,2-*b*]thiophene-5-carbaldehyde **5**: orange solid; mp: 105–106 °C; 58 mg, 0.08 mmol, 66%; R_f_ = 0.7 (pentane/CH_2_Cl_2_/Et_3_N = 2/1/0.02). ^1^H-NMR (400 MHz, CDCl_3_) δ (ppm): 0.90–0.93 (overlapped peaks, 12H), 1.33–1.37 (overlapped peaks, 16H), 1.45–1.56 (overlapped peaks, 8H), 1.81–1.94 (overlapped peaks, 8H), 4.08 (t, 2H, *J* = 6.5 Hz), 4.32 (t, 2H, *J* = 6.5 Hz), 4.42 (t, 2H, *J* = 6.5 Hz), 4.54 (t, 2H, *J* = 6.5 Hz), 6.26 (s, 1H), 10.00 (s, 1H). ^13^C-NMR (100 MHz, CDCl_3_) δ (ppm): 14.15, 14.20, 14.22, 22.70, 22.73, 22.79, 25.61, 25.68, 25.80, 25.86, 29.17, 29.96, 30.13, 30.15, 31.56, 31.65, 31.68, 31.72, 70.91, 72.70, 72.82, 73.07, 98.82, 115.96, 122.61, 124.29, 126.52, 126.59, 130.05, 135.90, 144.48, 146.61, 150.48, 157.96, 181.29. HRMS (ESI): *m*/*z* calcd for [C_37_H_54_O_5_S_4_ + H]^+^: 707.2927; found: 707.2957.

3,3’,6,6’-tetrakis(hexyloxy)-2,2’-bithieno[3,2-*b*]thiophene-5,5’-dicarbaldehyde **6**: red solid; mp: 192–193 °C; 140 mg, 0.19 mmol, 86%; R_f_ = 0.5 (pentane/CH_2_Cl_2_/Et_3_N = 2/1/0.02). ^1^H-NMR (400 MHz, CDCl_3_) δ (ppm): 0.90–0.94 (overlapped peaks, 12H), 1.35–1.39 (overlapped peaks, 16H), 1.48–1.56 (overlapped peaks, 8H), 1.84–1.93 (overlapped peaks, 8H), 4.42 (t, 4H, *J* = 6.5 Hz), 4.55 (t, 4H, *J* = 6.5 Hz), 10.03 (s, 2H). ^13^C-NMR (100 MHz, CDCl_3_) δ (ppm): 14.15, 14.21, 22.70, 22.77, 25.60, 25.78, 29.96, 30.16, 31.56, 31.68, 72.93, 73.27, 123.56, 123.73, 126.29, 134.39, 146.16, 157.62, 181.53. HRMS (ESI): *m*/*z* calcd for [C_38_H_54_O_6_S_4_ + H]^+^: 735.2876; found: 735.2906.

2-((3,3’,6,6’-tetrakis(hexyloxy)-2,2’-bithieno[3,2-*b*]thiophen-5-yl)methylene)malononitrile **7**: metallic green solid; mp: 189–190 °C; 44 mg, 0.06 mmol, 82%; ^1^H-NMR (400 MHz, CDCl_3_) δ (ppm): 0.90–0.95 (overlapped peaks, 12H), 1.35–1.40 (overlapped peaks, 16H), 1.47–1.56 (overlapped peaks, 8H), 1.81–1.94 (overlapped peaks, 8H), 4.09 (t, 2H, *J* = 6.5 Hz), 4.32 (t, 2H, *J* = 6.5 Hz), 4.48 (t, 2H, *J* = 6.3 Hz), 4.58 (t, 2H, *J* = 6.3 Hz), 6.31 (s, 1H), 7.90 (s, 1H). ^13^C-NMR (100 MHz, CDCl_3_) δ (ppm): 14.17, 14.20, 14.23, 22.70, 22.71, 22.74, 22.80, 25.65, 25.68, 25.80, 25.88, 29.18, 29.91, 30.09, 30.15, 31.57, 31.68, 31.72, 68.51, 70.98, 72.81, 73.20, 73.34, 99.70, 115.11, 115.19, 116.36, 116.48, 122.64, 126.12, 129.86, 131.15, 137.91, 144.18, 146.41, 147.67, 150.46, 157.94. HRMS (APCI): *m*/*z* calcd for [C_40_H_54_N_2_O_4_S_4_ + H]^+^: 755.3039; found: 755.3036.

(*Z*)-3-ethyl-5-((3,3’,6,6’-tetrakis(hexyloxy)-2,2’-bithieno[3,2-*b*]thiophen-5-yl)methylene)-2-thioxothiazolidin-4-one **8**: metallic green solid; mp: 138–139 °C; 40 mg, 0.05 mmol, 55%; R_f_ = 0.65 (pentane/CH_2_Cl_2_/Et_3_N = 2/1/0.02). ^1^H-NMR (600 MHz, CD_2_Cl_2_) δ (ppm): 0.91–0.96 (overlapped peaks, 12H), 1.25 (t, 3H, *J* = 7.1 Hz), 1.36–1.41 (overlapped peaks, 16H), 1.47–1.52 (overlapped peaks, 4H), 1.55–1.62 (overlapped peaks, 4H), 1.81–1.95 (overlapped peaks, 8H), 4.07 (t, 2H, *J* = 7.0 Hz), 4.14 (q, 2H, *J* = 7.1 Hz), 4.35 (t, 2H, *J* = 6.5 Hz), 4.42 (t, 2H, *J* = 6.5 Hz), 4.52 (t, 2H, *J* = 6.5 Hz), 6.28 (s, 1H), 7.99 (s, 1H). ^13^C-NMR (150 MHz, CD_2_Cl_2_) δ (ppm): 12.56, 14.36, 14.38, 14.42, 14.45, 23.15, 23.17, 23.19, 23.21, 26.06, 26.17, 26.21, 26.30, 29.65, 30.46, 30.57, 30.60, 32.07, 32.14, 32.16, 32.17, 40.34, 71.42, 73.24, 73.43, 73.56, 99.37, 116.00, 116.29, 119.12, 123.90, 125.28, 125.55, 127.00, 130.24, 134.54, 144.71, 146.97, 150.81, 154.44, 167.49, 192.57. HRMS (APCI): *m*/*z* calcd for [C_42_H_59_NO_5_S_6_ + H]^+^: 850.2790; found: 850.2819.

(*Z*)-2-(3-oxo-2-((3,3’,6,6’-tetrakis(hexyloxy)-2,2’-bithieno[3,2-*b*]thiophen-5-yl)methylene)-2,3-dihydro-1*H*-inden-1-ylidene)malononitrile **9**: metallic green solid; mp: >360 °C; 25 mg, 0.03 mmol, 57%; R_f_ = 0.7 (pentane/CH_2_Cl_2_/Et_3_N = 4/5/0.04). ^1^H-NMR (600 MHz, CDCl_3_) δ (ppm): 0.92–0.97 (overlapped peaks, 12H), 1.38–1.43 (overlapped peaks, 16H), 1.47–1.54 (overlapped peaks, 4H), 1.59–1.64 (overlapped peaks, 4H), 1.85 (m, 2H), 1.91 (m, 2H), 1.93–1.99 (overlapped peaks, 4H), 4.04 (t, 2H, *J* = 6.4 Hz), 4.43 (t, 2H, *J* = 6.4 Hz), 4.48 (t, 2H, *J* = 6.4 Hz), 4.66 (t, 2H, *J* = 6.4 Hz), 6.21 (s, 1H), 7.45 (t, 1H, *J* = 7.4 Hz), 7.50 (t, 1H, *J* = 7.2 Hz), 7.73 (d, 1H, *J* = 7.1 Hz), 8.37 (d, 1H, *J* = 7.6 Hz), 9.00 (s, 1H). ^13^C-NMR (150 MHz, CDCl_3_) δ (ppm): 14.22, 14.25, 14.27, 14.28, 22.77, 22.82, 25.72, 25.85, 25.86, 25.93, 29.94, 29.86, 30.14, 30.26, 31.69, 31.73, 31.81, 31.85, 65.62, 70.88, 72.89, 73.07, 73.77, 99.80, 115.43, 115.63, 116.19, 117.51, 120.53, 122.55, 122.85, 124.29, 125.78, 131.17, 131.75, 133.04, 133.17, 133.85, 136.64, 140.01, 142.63, 144.75, 148.02, 150.37, 160.49, 161.33, 188.87. Elemental analysis: calcd. for C_49_H_58_N_2_O_5_S_4_: C 66.63, H 6.62, N 3.17, S 14.52; found: C 66.87, H 6.48, N 3.11, S 14.36.

2,2’-(3,3’,6,6’-tetrakis(hexyloxy)-2,2’-bithieno[3,2-*b*]thiophene-5,5’-diyl)bis(methan-1-yl-1-ylidene)dimalononitrile **10**: metallic green solid; mp: 311–312 °C; 65 mg, 0.08 mmol, 96%; ^1^H-NMR (600 MHz, CDCl_3_) δ (ppm): 0.90–0.95 (overlapped peaks, 12H), 1.33–1.40 (overlapped peaks, 16H), 1.48–1.52 (overlapped peaks, 4H), 1.56–1.59 (overlapped peaks, 4H), 1.87–1.93 (overlapped peaks, 8H), 4.47 (t, 4H, *J* = 6.4 Hz), 4.59 (t, 4H, *J* = 6.6 Hz), 8.00 (s, 2H). ^13^C-NMR (150 MHz, CDCl_3_) δ (ppm): 14.15, 14.19, 22.67, 22.75, 25.62, 25.82, 29.93, 30.13, 31.54, 31.70, 71.35, 73.47, 73.58, 114.56, 115.58, 117.87, 125.33, 125.36, 135.55, 146.65, 146.77, 157.53. HRMS (APCI): *m*/*z* calcd for [C_44_H_54_N_4_O_4_S_4_ + H]^+^: 831.3101; found: 831.3090.

(5*Z*,5’*Z*)-5,5’-(3,3’,6,6’-tetrakis(hexyloxy)-2,2’-bithieno[3,2-*b*]thiophene-5,5’-diyl)bis(methan-1-yl-1-ylidene)bis(3-ethyl-2-thioxothiazolidin-4-one) **11**: metallic green solid; mp: 249–250 °C; 135 mg, 0.132 mmol, 75%; R_f_ = 0.75 (toluene/CH_2_Cl_2_ = 2/0.2). ^1^H-NMR (600 MHz, CDCl_3_) δ (ppm): 0.92–0.96 (overlapped peaks, 12H), 1.28 (t, 6H, *J* = 7.1 Hz), 1.37–1.42 (overlapped peaks, 16H), 1.52 (m, 4H), 1.61 (m, 4H), 1.87 (m, 4H), 1.94 (m, 4H), 4.17 (q, 4H, *J* = 7.2 Hz), 4.42 (t, 4H, *J* = 6.4 Hz), 4.49 (t, 4H, *J* = 6.6 Hz), 8.03 (s, 2H). ^13^C-NMR (150 MHz, CDCl_3_) δ (ppm): 12.46, 14.19, 14.27, 22.72, 22.81, 25.66, 25.89, 30.08, 30.22, 31.64, 31.79, 40.05, 73.06, 73.15, 116.90, 120.09, 122.20, 123.42, 126.80, 132.70, 145.88, 153.61, 167.27, 191.89. HRMS (ESI): *m*/*z* calcd for C_48_H_64_N_2_O_6_S_8_: 1020.2525; found: 1020.2581.

## 4. Conclusions

In conclusion, a series of symmetrical and unsymmetrical conjugated systems based on a 2,2′-bi[3,2-*b*]thienothiophene donor block and various electron-withdrawing groups as acceptors was synthesized and characterized. An in-depth study of the terminal electron-acceptor unit’s impact on the optical and electrochemical properties was carried out. It was shown that the introduction of electron-withdrawing groups in parent 2,2′-bi[3,2-*b*]thienothiophene led to a red shift of the main absorption band due to a decrease in the HOMO-LUMO band gap. All investigated systems display emission maxima in the green region, with the **A-D-A** compounds exhibiting poorer photoluminescence properties compared to their **D-A** counterparts. Finally, theoretical studies shed light on the observed optoelectronic behaviours by revealing the type and nature of the transitions that can take place and also by showing how various electron-accepting groups tune the optical bandgaps.

## Data Availability

Not applicable.

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
