# Peer review of "Exploring the Optoelectronic Properties of D-A and A-D-A 2,2′-bi[3,2-*b*]thienothiophene Derivatives"

_molecules, 2022, doi:10.3390/molecules27238463_

Round 1
Reviewer 1 Report
The authors synthesized a series of symmetrical and unsymmetrical conjugated systems based on 2,2’-bi[3,2-b]thienothiophene donor block and different electron-withdrawing end group. The materials with different end groups demonstrates different optical and electrochemical properties. This work affords good guidance for A-D-A and D-A compounds. Therefore, I recommend this work to be published after minor revisions.
1. The dot lines in Figure 1 and Figure 2 are not clear enough.
2. How about the molecular stacking of these compounds? XRD or GIWAXS is suggest to be conducted to investigate the molecular stacking.
3. To help readers have a good understanding the application of TT derivatives in non-fullerene acceptors, the related reviews are suggested to be cited, e.g. Nat. Rev. Chem. 2022; 6: 614-634; Adv. Powder Mater. 2022; 1: 100045.
Author Response
Thank you for the valuable suggestions! Please find below answers to the presented queries and the changes were also highlighted in the text:
We have changed the lines in Figures 1 and 2 so that they are more visible to the reader
We are not able to record XRD or GIWAXS data for our compounds; we have refined the text containing reference to H- and J-aggregates formation from the spectra in film vs solution, which also relates to the way the molecules may pack
We have inserted the suggested reference.
Reviewer 2 Report
In the manuscript, the authors synthesized a series of [3,2-b]-TT derivatives. The optical, and electrochemical properties studies are routine. The authors demonstrate the strength of the acceptor unit affects band gap energies for these compounds. However, considering that there are several inconsistencies in the scientific content, this contribution is premature to be considered for publication. I would recommend that a revised manuscript may become acceptable for publication in Molecules after addressing some of the questions and comments outlined below.
1. Due to the presence of long alkyl chains, the integrations for H-NMR are inaccurate, the hexanes and/or other solvent residues in H-NMR spectra should be marked out.
2. Line 122-123, ‘These peculiar features could suggest the formation of J- and H-aggregates…’ Could you explain more regarding this content (differences in 9, 7,8,10,11)?
3. Line 139, ‘photo-induced electron transfer from the donor to acceptor unit’, under most of the situations, ‘photo-induced electron transfer’, i.e., PET, I recommend using intramolecular charge transfer.
4. Line 150-152, I am not convinced by the author's explanation for the low QY found in 8 and 11. This behavior should be justified by additional experimental evidence, not only by their own published literature.
a. The absorption and fluorescence emission spectra should be analyzed in solvents with various polarities to check if one could observe emission in less polar solvents.
b. lifetime for these compounds should be collected to estimate the radiative decay constant Kr and non-radiative decay constant Knr. Presumably, the low QYs in a medium polar solvent are attributed to the very small energy gap, the emitting is subjected to the ‘energy gap law’, and non-radiative decay prevailed.
The authors have some misunderstanding between the transition dipole (moment) µtr and dipole moment change Δµge in the ground state and excited state.
The larger µtr means a higher transition probability, which would result in large oscillator strength f and induced stronger absorption band. As you could see in Table S1, the transition moment dipole is always large, which is in good agreement with the strong low-energy absorption band. However, it is not directly related to emission efficiencies.
The dipole moment change Δµge in the ground state and excited state only can reflect the direction of charge transfer. For instance, Δµge for A-D-A type compounds 6, 10, 11 is almost 0, however, this would not related to its QY.
5. The computational details should be provided in the ESI in detail, the reviewer cannot figure out which excited state (S1?) is indicated.
6. Figure 5, please provide the isovalue for the FMOs plot. It is hard to see the charge-transfer effect based on the current spatial distribution.
7. Figure 4, the discussion of the optimized geometries of ground states will not give too much useful information. The author should pay attention to the structural geometries comparison of the S0 and the excited state.
Scientific languages:
Line 164, ‘wholly’ should be ‘fully’
Line 171, ‘tendency’ should be ‘trend’
Line 172, ‘…the increasing strength of the electron-withdrawing group…’ can be ‘ the more electron-withdrawing group’
Author Response
Thank you for the valuable suggestions, which improved significantly our manuscript and helped us understand better aspects of our compounds’ behaviour! Please find below answers to the presented queries; the changes were also highlighted in the text:
A1) we have marked the solvent residues in 1H NMR spectra
A2) we have refined the text containing reference to J- and H-aggregates formation from the shifting of bands in film vs solution spectra for 7-8 and 10-11 vs. 9.
A3, A6) The isovalue for spatial distribution of FMOs is 0.02. We changed this value in order to see if we can obtain a more intuitive distribution, but we realized that in the case of ADA molecules an internal charge transfer (ICT) indeed does not take place, the transitions being classified as local excitations. We also computed the natural transition orbitals (NTOs) and we obtained the same distribution as for canonical FMOs. This was somehow expected since the S0 →S1 transition involve only the HOMO →LUMO transition. Furthermore, we calculate the difference electron density by subtracting the electron density of ground state from the electron density of first excited state. We obtained a nice difference density distribution (introduced as figure in text) which suggests that in the case of ADA molecules, the S0 →S1 transition is not accompanied by ICT.
On the other hand, in the case of D-A molecules, S0 →S1 transition is a combination of ICT and local excitation. Both FMOs and difference electron densities show that the electron density decreased considerably on the uncapped TT moiety after excitation and increased on the acceptor moiety. However, local excitations also take place in the other half of TT core and also in the EWGs part. Thus, S0 →S1 transition in the case of D-A compounds can be regarded as a combination of ICT and local excitation. We have adjusted the discussion to explain the obtained results
A4) Our compounds are not soluble in less polar solvents, so we could not record their spectra in such conditions. We have recorded lifetime measurements and confirmed that the non-radiative decay prevails in compounds with indandione moiety and A-D-A systems as well, compounds with very low QY. We understand our confusion in dipole moments and we have removed that discussion.
A5, A7) We have introduced computational details in ESI and extended the discussion on the excited states. We perfectly agree that the geometries of ground states are not relevant for the type and nature of transitions that can take place. We restricted the discussion on the geometries of ground states and we only emphasized the planarity of the molecules. We have moved the figure containing the optimized geometries in ground state to ESI. Also, we did not optimize the geometries of excited states. The theoretical spectra considering only vertical transitions match well the experimental ones, thus we did not consider running a Franck-Codon analysis which call for the optimization and detailed discussion of excited states.
Round 2
Reviewer 2 Report
My questions and comments have all been addressed.
However, the author's scientific language/English still has some room to improve, the newly updated part seems to have lots of grammatical errors, for example:
1. Line 214-215;
2. Line 206, ' mention' should be ' note'
3. Line 199-201
It would be better to have extensive checking/editing for this manuscript.
Author Response
Thank you for your suggestions and for the work done with our manuscript! We have edited the language throughout the paper and especially in the newly introduced part and the readability has improved. The main changes to the aforementioned text are highlighted in the updated version of the manuscript.